# Genetic Characterization of Rat Hepatic Stellate Cell Line PAV-1

**DOI:** 10.3390/cells12121603

**Published:** 2023-06-11

**Authors:** Kiara Gäberlein, Sarah K. Schröder, Indrajit Nanda, Claus Steinlein, Thomas Haaf, Eva M. Buhl, Patrick Sauvant, Vincent Sapin, Armand Abergel, Ralf Weiskirchen

**Affiliations:** 1Institute of Human Genetics, Julius Maximilians University of Würzburg, D-97074 Würzburg, Germany; kiara.gaeberlein@stud-mail.uni-wuerzburg.de (K.G.); nanda@biozentrum.uni-wuerzburg.de (I.N.); claus.steinlein@biozentrum.uni-wuerzburg.de (C.S.); thomas.haaf@uni-wuerzburg.de (T.H.); 2Institute of Molecular Pathobiochemistry, Experimental Gene Therapy and Clinical Chemistry (IFMPEGKC), RWTH University Hospital Aachen, D-52074 Aachen, Germany; saschroeder@ukaachen.de; 3Electron Microscopy Facility, Institute of Pathology, RWTH Aachen University Hospital, D-52074 Aachen, Germany; ebuhl@ukaachen.de; 4UMR CNRS 5248, CBMN, University Bordeaux, 33600 Pessac, France; patrick.sauvant@agro-bordeaux.fr; 5Feed & Food Department, Bordeaux Sciences Agro, 33175 Gradignan, France; 6Team “Translational Approach to Epithelial Injury and Repair”, Institute Genetics, Reproduction and Development (iGReD), Université Clermont Auvergne, 63000 Clermont-Ferrand, France; vincent.sapin@uca.fr; 7Department of Digestive and Hepatobiliary Medecine, CHU Clermont-Ferrand, 63000 Clermont-Ferrand, France; aabergel@chu-clermontferrand.fr

**Keywords:** liver fibrosis, extracellular matrix, hepatic stellate cells, myofibroblasts, retinol, metabolism, SKY analysis, karyogram, collagen, α-smooth muscle actin

## Abstract

The rat hepatic stellate cell line PAV-1 was established two decades ago and proposed as a cellular model to study aspects of hepatic retinoic acid metabolism. This cell line exhibits a myofibroblast-like phenotype but also has the ability to store retinyl esters and synthesize retinoic acid from its precursor retinol. Importantly, when cultured with palmitic acid alone or in combination with retinol, the cells switch to a deactivated phenotype in which the proliferation and expression of profibrogenic marker genes are suppressed. Despite these interesting characteristics, the cell line has somehow fallen into oblivion. However, based on the fact that working with in vivo models is becoming increasingly complicated, genetically characterized established cell lines that mimic aspects of hepatic stellate cell biology are of fundamental value for biomedical research. To genetically characterize PAV-1 cells, we performed karyotype analysis using conventional chromosome analysis and multicolor spectral karyotyping (SKY), which allowed us to identify numerical and specific chromosomal alteration in PAV-1 cells. In addition, we used a panel of 31 species-specific allelic variant sites to define a unique short tandem repeat (STR) profile for this cell line and performed bulk mRNA-sequencing, showing that PAV-1 cells express an abundance of genes specific for the proposed myofibroblastic phenotype. Finally, we used Rhodamine-Phalloidin staining and electron microscopy analysis, which showed that PAV-1 cells contain a robust intracellular network of filamentous actin and process typical ultrastructural features of hepatic stellate cells.

## 1. Introduction

Immortalized hepatic stellate cell (HSC) lines are useful experimental tools for the study of cellular aspects of hepatic fibrogenesis [1,2]. Like primary HSC, most of these continuously growing cell lines produce a well-developed α-smooth muscle actin (α-SMA) network, express a multitude of characteristic connective tissue markers, and have the capacity to take up and esterify retinol [1,2]. Compared to their primary counterparts that have a limited lifespan, established HSC lines grow faster and can be continuously passaged, indicating that these cells have developed features that allow them to escape from senescence. Consequently, these cell lines can be grown easily in almost infinite quantities making them attractive for many cellular and biochemical studies. Moreover, several HSC lines are a collection of cells originating from clonal expansion, suggesting that the genetic variation in this cell pool is minimal compared to primary HSC, in which subsets of cells exist in which the gene expression can vary. In particular, the heterogeneity of primary cells in different species became evident in cell tracing and single-cell RNA sequencing experiments [3,4,5,6,7,8]. The feature that cell lines are more or less genetically characterized enables researchers to replicate experiments more easily. Over the years, immortalized cell lines have increasingly emerged as a suitable alternative to primary HSCs, partly due to the fact that the overall isolation of primary cells is not only time-consuming and involves huge costs for implementation and reagents [1] but also requires critical consideration regarding the use of animals in the context of the 3R concept (Refine, Reduce, Replace) [9]. However, there are also some caveats when working with immortalized cell lines. Most importantly, the majority of HSC lines are derived from primary HSC that were either transformed with the Simian virus 40 large T-antigen, immortalized by introducing telomerase reverse transcriptase, or immortalized by exposure to intensive radiation with ultraviolet light, while there are only some lines that became spontaneously immortalized in culture without further treatment [10]. As such, most of the cell lines are genetically modified organisms (GMOs) in which the DNA has been altered using genetic engineering techniques. In addition, as with other cell lines, prolonged passaging or inter-laboratory effects can produce genomic instability and genetic drift, which, in the long-term, might provoke alterations in morphological features, proliferative capacity, gene expression patterns, and responses to proteotoxic stress and drugs [11]. Another critical issue when working with cell lines is the occurrence of cell line cross-contamination and misidentification, which significantly contributes to the growing concerns about errors, false conclusions, and irreproducible results in biomedical research [12]. For example, GREF-X cells were originally described as a human liver myofibroblast cell line established from explants of human liver that were transfected with a plasmid encoding polyoma virus large T antigen [13]. However, two years later it was found that this line was of rat origin and listed by the International Cell Line Authentication Committee (ICLAC) under Registration ID: ICLAC-00123 as a misidentified cell line [14,15]. Cell line misidentification is highly frequent and the portion of misidentified cell lines can thus only be uncovered by implementing a strict mandatory cell line authentication policy [16]. In particular, the establishment of new cell lines is highly challenging because cross-contamination at the source is one of the most common causes for the development of misidentified cell lines [14,17]. However, to identify a cell line, it is necessary to urgently define specific biological markers and characteristics that allow for the differentiation of the line from all others. Regarding HSC lines, this can be performed in the simplest way via the expression of specific transgenes (SV40T, hTERT, GFP) or typical HSC markers (collagen type I, α-SMA, GFAP, desmin). Nowadays, karyotype analysis (which can detect gross abnormalities in the number and structure of chromosomes), single nucleotide polymorphisms (SNP) profiling (which allows for the identification of mutations), screening with species-specific primers, whole genome sequencing (WGS), and CO1 barcoding (which determines sequences within the cytochrome c oxidase subunit 1 (CO1)) have been applied to cell line authentication [17]. The most preferred method for cell line authentication is short tandem repeat (STR) profiling, which has high discriminatory power [17].

In previous studies, we identified several genetic characteristics and STR profiles for several HSC lines, including human LX-2 [18], rat cell lines HSC-T6 and CFSC-2G [19,20], and mouse HSC cell lines GRX [21] and Col-GFP HSC [22]. Now we have extended our studies and performed karyotype analysis and multicolor spectral karyotyping (SKY) in the rat HSC line PAV-1, which was isolated by amplification of a colony obtained via the spontaneous immortalization of primary HSC isolated from an over 8-month-old male Wistar rat [23]. In our study, we have identified unique chromosomal characteristics for PAV-1 cells by using conventional chromosome analysis of metaphase spreads. Furthermore, several cryptic chromosome rearrangements were detected with spectral karyotyping.

Moreover, we used a panel of 31 species-specific allelic variant sites to define a unique STR profile for PAV-1 cells. PAV-1 cells cultured under basal conditions were further subjected to bulk mRNA-sequencing, showing that PAV-1 cells express an abundance of genes specific for HSC. By employing Rhodamine-Phalloidin staining, we further demonstrate that PAV-1 cells form a robust intracellular network of filamentous actin. PAV-1 cells further possess ultrastructural features of HSC, including large, electron-dense nuclei with excessive bulges of the nuclear envelope, elongated mitochondria, distinct rough endoplasmic reticulum, pronounced lysosomes, and an abundance of intracellular lipid droplets, as assessed by electron microscopy analysis.

In summary, this study defines unique authentication standards for PAV-1 cells that will form the basis for avoiding misidentification when using this cell line in biomedical research.

## 2. Materials and Methods

### 2.1. Culturing of PAV-1 Cells

PAV-1 cells were initially established and characterized two decades ago by Patrick Sauvant and colleagues [23,24]. The cells were spontaneously immortalized from cultured primary rat HSCs that were purified from male Wistar rats by a pronase-collagenase perfusion protocol [23]. In our laboratory, PAV-1 cells were maintained in Dulbecco’s modified Eagle’s medium (DMEM) (#D6171), supplemented with 10% fetal bovine serum (FBS, #F7524), 2 mM L-Glutamine (#G7513), 1 mM sodium pyruvate (#S8636), and 1× Penicillin/Streptomycin (#P0781). All reagents were obtained from Sigma-Aldrich (Taufkirchen, Germany). For routine culturing, the medium was replaced every third day and cells were sub-cultured using Accutase^®^ solution (#A6964, Sigma-Aldrich). Detection of potential *Mycoplasma* spp. contaminations in cell culture supernatants was completed using the Venor^®^GeM OneStep kit (#11-8050, Minerva biolabs GmbH, Berlin, Germany) according to the manufacturer’s instructions. The PCR products were separated on a 2% standard agarose gel including ethidium bromide and amplicons visualized using a standard gel imager. The clearance of PAV-1 cells from Mycoplasma contamination was completed using the Plasmocin^®^-Mycoplasma elimination reagent and by essentially following the instructions provided by the manufacturer (abt-mpt-1, InvivoGen, Toulouse, France). The rat HSC line HSC-T6 and murine hepatocytic AML-12 cells were cultured as previously described [19,21].

### 2.2. Electron Microscopy Analysis

Electron microscopy analysis of PAV-1 cells was completed as described previously for the rat HSC line HSC-T6 [19]. In brief, cells were fixed, dehydrated, and subsequently embedded in Epon resin. Ultrathin sections (70–100 nm) were prepared and packed upon Cu/Rh grids and stained with 0.5% uranyl acetate and 1% lead citrate for better contrast. The samples were analyzed at an acceleration voltage of 60 kV using a Zeiss Leo 906 (Carl Zeiss AG, Oberkochen, Germany) transmission electron microscope. The magnification range varied from 4646× to 27,800×.

### 2.3. Preparation of PAV-1 Metaphase Chromosomes and Karyotyping

Chromosomes of PAV-1 cells were prepared according to standard protocols [19]. Briefly, cultures of semi-confluent PAV-1 cells were exposed to colcemid solution (Gibco, ThermoFisher Scientific, Dreieich, Germany), detached through brief trypsinization, harvested by centrifugation, treated with 0.56% (*w*/*v*) hypotonic potassium chloride solution, and fixed with a mixture of cold methanol and acetic acid (3:1) at 37 °C. For G-banding, air-dried chromosome spreads were prepared, and slides were treated with 0.025% (*w*/*v*) trypsin solution followed by staining with Giemsa solution. The heterochromatin in metaphase chromosomes was stained using a standardized protocol [25]. At least 10 GTG-banded metaphases from different culture passages were photographed and karyotyped according the method outlined in [26].

### 2.4. In Situ Hybrdization, Spectral Imaging, and Nucleolar Organizer Region Staining

Detection of chromosome rearrangements in PAV-1 cells was accomplished using a commercially available rat SKY probe (Applied Spectral Imaging Inc., Carlsbad, CA, USA) and published protocols [19,20,27]. For silver (Ag)-staining of the nucleolus organizer regions (NORs), slides with metaphase chromosomes were stained with AgNO_3_ according to the method of Goodpasture and Bloom [28].

### 2.5. Short Tandem Repeat (STR) Profiling

STR profiling and interspecies contamination testing for PAV-1 cells was performed using the commercially offered cell line authentication service from IDEXX (Kornwestheim, Germany). The profiling was completed using the CellCheck^TM^ Rat system that includes 31 species-specific STR markers that are distributed on the 20 different rat autosomes. The mouse cell line AML12 was subjected for STR profiling using the CellCheck^TM^ Mouse system that contains the 19 species-specific consensus STR markers proposed by the Consortium for Mouse Cell line Authentication for STR profiling in mouse [29,30]. An STR similarity search with the obtained STR profile of AML12 was performed using the Cellosaurus STR Similarity Search Tool CLASTR 1.4.4 (release 41.0) and the Cellosaurus mouse STR database [31,32]. In the search, the settings were set to the following: Scoring algorithm: Tanabe, Mode: Non-empty markers, Score filter: 40%, and Min. Markers: 8.

### 2.6. Next Generation Sequencing and Data Analysis

RNA from cultured PAV-1 grown to a density of 80% was isolated by a standard CsCl density gradient centrifugation as described before [19]. The resulting RNA pellet was re-suspended in sterile water, purified by ethanol precipitation, re-suspended in sterile water, and quantified by using UV spectroscopy. For RNA bulk sequencing, the quality of the RNA was determined on the Agilent 4200 TapeStation platform (Agilent Technologies Inc., Waldbronn, Germany). Depletion of ribosomal RNAs, library preparation, sequencing, and bioinformatics analysis were essentially performed as described before [19]. The abundance of found individual gene transcripts is given in ‘Transcripts Per Million (TPM)’.

### 2.7. Conventional Reverse Transcription Polymerase Chain Reaction

For conventional Reverse Transcription Polymerase Chain Reaction (RT-PCR), cDNA was synthesized from RNA as previously described [21]. The cDNA was subjected to the following cycle conditions: 5 min initial denaturation at 95 °C, 1 min at 95 °C, 1 min annealing at 60 °C (35 cycles), 3 min extension at 72 °C, and final elongation at 72 °C for 10 min. The expected sizes of amplicons generated by the chosen primer combinations were 94 bp for rat collagen type I alpha 1 (r*Col1a*, NM_053304.1, for: 5′-cat gtt cag ctt tgt gga cct-3′, rev: 5′-gca gct gac ttc agg gat ct-3′), 73 bp for rat fibronectin (r*Fn1*, NM_019143.2, for: 5′-cag ccc ctg att gga gtc-3′, rev: 5′-tgg gtg aca cct gag tga ac-3′), 68 bp for rat glial fibrillary acidic protein (r*Gfap*, NM_017009.2, for: 5′-ttt ctc caa cct cca gat cc-3′, rev: 5′-tct tga ggt ggc ctt ctg ac-3′), and 111 bp for rat glyceraldehyde 3-phosphate dehydrogenase (r*Gapdh,* NM_017008.4, for: 5′-aac ctg cca agt atg atg aca tca-3′, rev: 5′-tgt tga agt cac agg aga caa cct-3′), respectively. Amplified PCR products were separated in a 2% agarose gel containing ethidium bromide in TAE buffer using gel imager Gel iX20 (Intas Science Imaging Instruments GmbH, Göttingen, Germany) and the GelPilot 100 bp ladder (#239045; Qiagen, Hilden, Germany) as a marker. For real-time quantitative PCR (RT-qPCR) the primers used for the amplification of *Gfap* and housekeeper gene *Gapdh* were subjected to a TaqMan analysis following standard protocols [33].

### 2.8. Western Blot Analysis

Preparation of protein extracts, protein quantification, and Western blot analysis were performed using established protocols [34]. In brief, equal protein amounts (40 µg/lane) were heated at 80 °C for 10 min and separated in 4–12% Bis-Tris gels (Invitrogen, Darmstadt, Germany) under decreasing conditions using MES running buffer. The separated proteins were then electro-blotted on nitrocellulose membranes (Schleicher & Schuell, Dassel, Germany), and successful protein transfer and equal protein loading controlled by Ponceau S stain. After blocking unspecific binding sites in 5% milk powder in Tris-buffered saline with Tween 20 (20 mM Tris, 150 mM NaCl, 0.1% (*w*/*v*) Tween 20 detergent), the membranes were successively probed with the primary antibodies depicted in Table 1. Primary antibodies were detected with horseradish peroxidase (HRP)-conjugated secondary antibodies and the Supersignal™ chemiluminescent substrate (Perbio Science, Bonn, Germany).

### 2.9. Rhodamine-Phalloidin Stain

A Rhodamine-Phalloidin-conjugate (#R415, Thermo Fisher Scientific) was used to stain the microfilaments in PAV-1 cells according to a previously established protocol [19,35]. The nuclei were counterstained by incubation in a 4′,6-diamidino-2-phenylindole (DAPI) solution (#D1306, Thermo Fisher Scientific), and the glass coverslips were mounted with PermaFluor aqueous mounting medium (#TA-030-FM, Thermo Fisher Scientific). Images of stained cells were captured using a Nikon Eclipse E80i fluorescence microscope (Nikon Europe, Düsseldorf, Germany) equipped with the NIS-Elements Vis software (Nikon Europe, version 3.22.01).

### 2.10. Lipid Droplet Staining

PAV-1 cells were stimulated with oleic acid (#O3008, Sigma-Aldrich) for 24 h and intracellular lipid droplets were stained with BODIPY^TM^ 493/503 (#D3922, Thermo Fisher Scientific) as described before [19]. The stained cells were analyzed using a Nikon Eclipse E80i microscope.

## 3. Results

### 3.1. Phenotypic Characterization of PAV-1 Cells

PAV-1 cells were established over two decades ago [23]. After receiving the cells from the original laboratory that generated this cell line, the cells were first stored in our laboratory for about 10 years in a liquid nitrogen tank and later transferred into a −150 °C freezer. Because we have not worked with this cell line before, we first tested for mycoplasma infection that might be introduced during establishment, handling, parceling, storing, or by contaminated cell culture reagents such as newborn bovine serum frequently contaminated with mycoplasma [36,37]. According to the information provided by the manufacturer, the conventional PCR-based assay used for this analysis is designed to specifically target and amplify the highly conserved 16S rRNA coding region of the mycoplasma genome, presumably allowing the detection of 85 different *Mycoplasma* species, 7 *Acholeplasma*, and 1 *Ureaplasma* including *M. orale*, *M. hyorhinis*, *M. arginini*, *M. fermentans*, *M. salivarium*, *M. hominis*, usually encountered as contaminants in cell cultures. Unfortunately, the original stock tested positive for mycoplasma infection; however, this could be alleviated via the usage of a Mycoplasma elimination reagent before starting with the phenotypic and genotypic characterization described in the following. Because we have not worked with this cell line before, we first tested for mycoplasma infection that might be introduced during establishment, handling, parceling, storing, or by contaminated cell culture reagents such as newborn bovine serum (frequently contaminated with mycoplasma; see Section 2.1) (Appendix A).

As already mentioned above, the PAV-1 cell line is a spontaneously immortalized HSC line derived from cultured primary rat HSCs originating from male Wistar rats [23]. These cells resemble activated HSC and share many characteristics with primary HSC/myofibroblasts. In particular, they acquire a distinct fibroblast-like morphology in culture, supporting the notion that they originate from connective tissue (Figure 1). At low density, PAV-1 cells appear in fusiform, elongated spindle-shaped structures, whereas at higher density, the cells form a uniform, dense cell layer.

Electron microscopy revealed that PAV-1 cells possess ultrastructural features that are hallmarks of cells originating from HSC, including large, electron-dense nuclei, elongated mitochondria, distinct rough endoplasmic reticulum, pronounced lysosomes, and, most importantly, a large abundance of intracellular lipid droplets (Figure 2).

Furthermore, in agreement with the proposed myofibroblastic phenotype, we could show that these cells form a robust network of cytoplasmic microfilaments, as assessed by Rhodamine-Phalloidin staining (Figure 3).

HSC typically contain large quantities of triglycerides, cholesterol ester, cholesterol, phospholipids, and free fatty acids that are stored in lipid droplets [38]. Importantly, the size and number of these lipid-storing vesicles is strongly dependent on dietary fatty acid intake, suggesting that these fat-storing cells have the capacity to uptake respective compounds [39]. The PAV-1 line was introduced as an activated rat HSC line with the ability to uptake, store, and metabolize fatty acids and vitamin A, thereby reversing the activated phenotype of PAV-1 into a quiescent phenotype [40]. In agreement with these findings, we demonstrated that the incubation of cultured PAV-1 with oleic acid (250 µM) resulted in the enlargement of lipid droplets, as demonstrated by BODIPY staining (Figure 4).

### 3.2. Expression of Hepatic Stellate Cell Markers in PAV-1

#### 3.2.1. Western Blot Analysis and Reverse Transcriptase PCR

We subsequently performed Western blot analysis for typical HSC markers using the rat HSC line HSC-T6 as a control. This analysis showed that PAV-1 cells express collagen type I, desmin, caveolin-1, vimentin, fibronectin, collagen type IV, and GFAP (Figure 5).

When compared to HSC-T6, which represents another immortalized HSC line, the expression of all these markers at protein level was significantly higher in PAV-1 cells. Only the expression of collagen type IV was slightly reduced in PAV-1 cells. As expected, the expression of the Simian virus large T-antigen (SV40T) was absent in PAV-1 cells, while the expression of SV40T was high in HSC-T6 that were immortalized with this dominant-acting oncogene. In addition, PAV-1 cells were negative for the nuclear receptor hepatocyte nuclear factor 1α (HNF-4α), representing a transcription factor that regulates the expression of several hepatocyte-specific genes during hepatic differentiation and proliferation [41]. Interestingly, AML12 cells that were used as a control for HNF-4α expression showed a faint expression of GFAP—an astrocyte marker that, within the liver, is typically only expressed in HSC and generally suggested as an early marker of stellate cell activation [42,43]. The respective band was found with two different antibodies. To document the identity and rule out cross-contamination with other cells in our AML12 culture that was used as a control, we performed STR profiling in AML12 using the CellCheck^TM^ Mouse system, including 19 species-specific STR markers, from which 18 are also implemented in the Cellosaurus cell line knowledge resource [31,32]. This analysis revealed that the AML12 cells used in our study had a unique STR profile and were not contaminated by other cells (Appendix A, Appendix A). The highest similarities regarding the 18 Cellosaurus markers were assigned to CVCL_0120 (3T-Swiss albino, 46.81%), CVCL_B6Z5 (MFC/HL-041, 44.44%), CVCL_0321 (HT22, 42.31%), CVCL_3420 (STO, 41.03%), CVCL_VR92 (MEC1, 40.82) and CVCL_ZL25 (GN11, 40.00%), respectively.

The expression of collagen type I (*Col1α*) and fibronectin (*Fn1*) was also confirmed by conventional reverse transcription polymerase chain reaction (RT-PCR) (Figure 6A). However, in both PAV-1 and HSC-T6, we found only low mRNA quantities of *Gfap* that were about 64-fold lower in PAV-1 than in HSC-T6, as indicated by the different C_t_ values in RT-qPCR that differed in their cycle threshold values by a factor of 2^6^ (Figure 6B). This finding contrasted the GFAP protein expression levels that were much higher in PAV-1 cells.

#### 3.2.2. Transcriptomic Analysis of PAV-1 Cells

To measure the general expression of individual genes in PAV-1 cells, we performed bulk RNA-sequencing (mRNA-seq). In the respective analyses, we showed that PAV-1 cells express a total of 23765 different mRNA species (Appendix A). The five highest expressions were found for ribonuclease pancreatic beta-type (*LOC103690354*, 17612.5 TPM), ferritin heavy chain 1 (*Fth1*, 17487.4 TPM), eukaryotic translation elongation factor 1-α1 (*Eef1a1*, 8875.58 TPM), ferritin light chain 1-like (*LOC100360087*, 7138.67 TPM), and vimentin (*Vim*, 7038.71), respectively. Importantly, PAV-1 cells expressed a large number of HSC markers that were also found in the rat HSC lines HSC-T6 and CFSC-2G (Table 2). Interestingly, the bulk mRNA sequencing confirmed our RT-PCR results, showing that *Gfap* mRNA is expressed in HSC-T6 at a rate that is about 60-fold higher than in PAV-1 (0.60103 TPM vs. 0.0115619).

Moreover, in line with previous reports and other rat HSC lines, we found that PAV-1 cell express many metabolic and nuclear retinoic acid receptors, including retinoic acid receptor (RAR)α, RARβ, RARγ, retinoid X receptor (RXR)α, RXRβ, retinol binding proteins, retinal dehydrogenases, and aldehyde dehydrogenases (Table 3, Appendix A).

Importantly, PAV-1 cells also express many genes that are involved in the building, storage, and hydrolysis of retinyl esters and genes with importance for Vitamin A metabolism and signaling. These include many members of the aldehyde dehydrogenase family (*Aldh1a1*, *Aldh1a2*, *Aldh1a3*, *Aldh1a7*, *Aldh1I1*, *Aldh1I2*, *Aldh2*, *Aldh3a1*, *Aldh3a2*, *Aldh3b1*, *Aldh5a1*, *Aldh6a1*, *Aldh7a1*, *Aldh8a1*, *Aldh9a1*, *Aldh16a1*, *Aldh18a1*), retinol dehydrogenases (*Rdh5*, *Rdh8*, *Rdh10*, *Rdh11*, *Rdh13*, *Rdh14*) and cellular retinoic acid binding proteins (*Crabp1*, *Crabp2*) (Appendix A). Moreover, genes that are key factors in β-carotene metabolism to vitamin A, such as the β-carotene oxygenases (*Bco1*, *Bco2*), and genes involved in bidirectional transport of Vitamin A between extra- and intracellular retinol binding proteins (RBPs), such as *Stra6* and *Stra8*, are expressed in PAV-1 (Appendix A).

### 3.3. Karyotype Analysis and Spectral Karyotyping of PAV-1 Cells

The counting of numerous Giemsa-stained metaphase spreads from different passages showed that the overall diploid chromosome numbers vary between 38 and 45, with most metaphases (>70%) displaying 43 chromosomes. However, in each culture passage, some cells were endowed with >80 chromosomes (nearly tetraploid, Appendix A). Interestingly, the number of tetraploid cells tends to outnumber the diploid cells in higher culture passages (Table 4). This phenomenon might be analogous to the increasing rate of genomic rearrangements in liver with age [44].

In each cell, a large metacentric and two small marker chromosomes were noted, which appear to be a typical feature of PAV-1 cells. The GTG banding pattern allowed us to classify the individual chromosomes and revealed several unique features of the PAV-1 karyotype. In GTG-banded karyotypes, the large metacentric chromosome was identified as an iso-chromosome of RNO5 (Figure 7A). The other notable feature of the karyotype is the presence of three copies of RNO7 (trisomy) and one copy of RNO12 (monosomy). Additionally, in one of the homologs of RNO4, an extra chromosomal segment at the terminal region of the long arm can be noted.

The C-banded metaphases unequivocally identified the heterochromatic Y-chromosome (Figure 7B), confirming that the investigated PAV-1 cell line was originally derived from a male rat. The remaining constitutively heterochromatic regions (C-bands) were confined to the centromeric regions of most chromosomes.

Figure 8 shows the silver staining of metaphase chromosomes, demonstrating six active nucleolar organizer sites (NORs). As expected, these NORs were located near the centromeres of RNO3, RNO11, and RNO12. Interestingly, the larger marker chromosomes had active NORs in each metaphase analyzed, suggesting that either an NOR was transferred to the marker chromosome, or alternatively, that the marker chromosome was derived from one of the three NOR-carrying rat chromosomes, RNO3, RNO11, or RNO12.

### 3.4. Spectral Karyotype Analysis

SKY analysis is based on specific color codes assigned to each chromosome and allows for the accurate identification of both numerical and structural rearrangements. Figure 9 displays a PAV-1 metaphase spread and representative SKY karyotype with chromosome-specific paintings.

In addition to the iso-chromosome of RNO5 and trisomy 7, there were three different chromosomes (RNO4, RNO11 and RNO12) containing segments of other chromosomes. For example, the spectral patterns of RNO19 and RNO20 were present on one homolog of RNO11 and RNO12—consistent with translocations between RNO11 and RNO12 and between RNO12 and RNO19, respectively. The second homologs of RNO19 and RNO20 maintained their morphology as individual chromosome units in the karyotype. Additionally, one of the homologs of RNO4 has a tiny segment from RNO11 at the terminal end of its long arm, implying that RNO11 material was translocated to RNO4. Based on its spectral pattern, the larger marker chromosome carrying a NOR represents duplicated material from RNO11. The spectral pattern of the smaller marker chromosome could not be assigned to a specific chromosome. The copy number changes of RNO5 and RNO7, as well as the above-mentioned structural changes in the diploid karyotype, were observed to occur twice in the tetraploid karyotype (Appendix A). In a few tetraploid metaphases, additional sporadic translocations were also noted. Both common and sporadic structural changes in diploid and tetraploid metaphases are depicted in Table 5. Sporadic structural changes were detected in less than 50% of cells.

### 3.5. Short Tandem Repeat Profiling

To establish a characteristic STR profile of PAV-1 cells, we genotyped 31 established polymorphic markers that we previously used to profile two other rat HSC lines, namely HSC-T6 [19] and CFSC-2G [20]. The sum of generated STR profiles for PAV-1 cells revealed a unique pattern of allele sizes for the 31 markers that was different from those of CFSC-2G and HSC-T6 that we have recently established (Table 6, Appendix A). This analysis confirmed that the PAV-1 cell line is rat-derived and is free from mammalian interspecies contamination.

## 4. Discussion

The authentication of a novel immortalized cell line for research and clinical use is mandatory when using a cell line in biomedical research [45]. Cell misidentification and cross-contamination are fatal issues that lead to error-filled publications, false data, irreproducible results, and a substantial waste of money [12,46]. By using complementary search strategies, Horbach and Halffman identified 32,755 articles in 2017 that reported research results with misidentified cells that, in turn, were cited by approximately half a million other papers [12]. Moreover, there are estimates that 16.1% of all published papers use problematic cell lines that are either contaminated or misidentified [47]. Other estimates even suggest that misidentified cell lines in biomedical research can be close to 50% in some areas of the world [48,49]. In particular, experts in this field estimate that one-third of all human cell lines are thought to be misidentified [46].

In liver research, there are many examples of misidentified cell lines, including Chang liver cells, GREF-X cells, LO2, WRL 68, and many others [10]. Nevertheless, continuously growing cell lines are still useful model systems for modern medical research because they provide an indefinite source of biological material for experiments [50]. In addition, well-characterized cell lines have the potential to replace some animal experiments, thereby fostering the ethical 3R (Replacement, Reduction, Refinement) framework proposed by William M. S. Russell and Rex L. Burch in 1959 [9]. Nowadays, many journals and funding agencies, including the National Institutes of Health (NIH), request that the cells used in experiments be subjected to authentication testing prior to publication or before providing funding or grants.

In the past, we determined genetic details of different continuous HSC lines, including human LX-2 [18], mouse cell lines GRX [21] and Col-GFP HSC [22], and rat lines HSC-T6 [19] and CFSC-2G [20]. Herein, we extended these studies and characterized the rat HSC line PAV-1 in regard to its genetic characteristics. This cell line was originally described as a convenient model to study aspects of vitamin A metabolism in HSC that can produce functional retinoids from retinol [23,51,52,53]. Moreover, a subsequent study found that treatment with palmitic acid alone or in combination with retinol significantly decreased cell proliferation and α-SMA expression, suggesting that PAV-1 cells might be suitable to study processes of HSC deactivation [40].

PAV-1 cells were originally introduced as an immortalized HSC line model with the capacity to convert retinol into retinoid acid [23]. The bulk mRNA-sequencing performed in our study confirmed that PAV-1 cells express important genes implicated in the building, storage, and hydrolysis of retinyl esters. Most important is the expression of retinoid acid receptors and retinol-binding proteins. Furthermore, important key genes that mediate the breakdown of β-carotene metabolism to vitamin A and genes involved in the bidirectional transport of vitamin A between extra- and intracellular RBPs are expressed in PAV-1. For example, *Bco1* is expressed in PAV-1 and is the only vitamin A-producing enzyme mediating the central cleavage across the C15,C15′ double bond adjacent to a canonical β-ionone ring site of carotenoids and β-apocarotenoids [54]. The expression of all these genes strongly confirms the notion that PAV-1 cells are an ideal experimental tool that can be used to study retinol metabolism.

In our study, we found that PAV-1 cells express GFAP both at mRNA and protein level. Although the expression of *Gfap* at mRNA level was extremely low, as assessed by the high C_t_ value in RT-qPCR, Western blot analysis showed that the cells expressed GFAP protein quantities that were comparable to that found in HSC-T6. In contrast, in the initial study describing the establishment of PAV-1, the cells were found to be negative for GFAP in immunohistochemistry [23]. In the previous study, the authors used a rabbit polyclonal GFAP antibody (#G9269, Sigma-Adrich, St._Quentin Fallavier, France) directed against rat and human GFAP, while, in the present study, we used another antibody (#ab7779, Abcam, Berlin, Germany) that detected a protein that was ~55 kDa in size and further confirmed with another antibody (#sc-33673, Santa Cruz Biotech., Santa Cruz, CA, USA). The overall reactivity of the antibody that was used in the previous study was significantly lower, preventing the identification of GFAP in immunocytochemistry. In this context it is worth noting that the rat gene *Gfap* produces several GFAP splice variants (GFAPα, GFAPβ, GFAPδ, GFAPε, GFAPκ). The difference between the splice variants lies either in the 5′-untranslated region in the case of GFAPα, GFAPβ, and GFAPγ or the 3′-untranslated region for GFAPδ, GFAPε, and GFAPκ [55,56]. Importantly, quiescent and culture-activated primary rat HSC GFAPα is the predominant form, while GFAPβ predominates in the SV40-immortalized cell line HSC-T6 [56]. All splice variants have a different half-life and are differentially regulated on a transcriptional level [56]. In our expression analysis, we used primers that bound to regions located in exons 7 and 8. With these primers, it is possible to amplify the transcripts GFAPα and GFAPβ, while the three other *Gfap* mRNAs that lack exons VIII are not amplified. Potentially, the inconsistency is based on the fact that, in PAV-1 cells, one of the other (GFAPδ, GFAPε, GFAPκ) mRNAs that are not amplified by our PCR strategy is expressed at higher levels than in HSC-T6.

Unlike the two previously published karyotypes of HSC lines, HSC-T6 and CFSC-2G, PAV-1 is an authentic spontaneously transformed hepatic cell line. In the current study, the application of conventional banding methods and SKY analysis was able to reveal several striking chromosome alterations in the PAV-1 cell line that include iso-chromosomes of RNO5, trisomy RNO7, monosomy RNO12, partial trisomy of RNO11, two marker chromosomes (one derived from RNO11), and cryptic translocations to RNO4, RNO11, and RNO12. At least four different chromosome breakages and reunions must have occurred during the formation of iso-chromosomes of RNO5 and the translocations involving RNO4, RNO11, and RNO12. However, due to technical limitations, small deletions, duplications, and inversions cannot be excluded by SKY analysis. The structural rearrangements identified in this study on RNO4, RNO11, and RNO12 appear to be specific to PAV-1 and are not observed in the karyotype of two HSC lines previously analyzed [19,20]. Interestingly, the gain of one copy of RNO7 and the loss of one copy of RNO12 are also noted in the previously analyzed HSC-T6 but not in CFSC-2G. Therefore, the analysis of different HSC lines can provide information on the recurrence of RNO7 trisomy and monosomy 12 during transformation. We have not addressed the impact of these chromosomal alterations on cellular features. However, it should be noted that the cell lines PAV-1 and HSC-T6 cells grow rather fast, while CFSC-2G cells grow significantly slower and require non-essential amino acids for optimal growth. Whether there is a link between the gain of one copy of RNO7 and the loss of one copy of RNO12 needs to be investigated in future studies. In particular, it will be essential to identify genes affected by these alterations and to clarify if this is only a coincidence in PAV-1 and HSC-T6 or if these changes are a prerequisite for the immortalized phenotype of these two cell lines.

In our study, we further analyzed the expression of typical HSC markers, showing that the PAV-1 cell line is capable of expressing collagen type I, desmin, caveolin-1, α-smooth muscle actin (α-SMA), vimentin, fibronectin, collagen type IV, and GFAP, while lacking the expression of the typical hepatocyte marker HNF-4α. Therefore, PAV-1 cells are suitable as an alternative HSC in vitro model for various scientific issues in contrast to isolated primary HSCs, which are associated with higher overall costs and always require the use of animals [1,9]. In addition, we showed that PAV-1 lacks the expression of SV40T, which is used to immortalize many other different HSC lines, including LX-1, SV68c-IS, A640-IS (33 °C), IMS/MT(−), IMS/N, HSC-T6, Col-GFP-HSC, JS1 (C57/Bl6 WT), JS2 (C57/Bl6 TLR4^−/−^), and JS3 (C57/Bl6 MyoD88^−/−^) [2,57,58,59,60,61,62,63].

PAV-1 cells were initially described as a spontaneous immortalized cell line. However, it might be possible that PAV-1 cells were immortalized by the activity of mycoplasma infection. It is well known that the persistent infection of cells with mycoplasma can lead to malignant transformations in cell culture and provoke prominent chromosomal changes [64,65]. In our study, we performed mycoplasma clearance for two weeks and showed that PAV-1 can be effectively propagated without the presence of mycoplasma, suggesting that this bacterial contaminant is not required to maintain the immortalized phenotype.

Mycoplasma contamination can exhibit negative effects on cell morphology, proliferation, gene expression, and responsiveness to stimuli [66]. We cannot rule out that the clearance of mycoplasma has affected the properties of the original PAV-1 cell. However, many other studies have unequivocally shown that the cytotoxic effects of antibiotics against *Mycoplasma* species are reversible. For example, Plasmocin treatment for five passages has not produced any obvious alterations in human embryonic stem cells [67].

The elimination of mycoplasma in PAV-1 culture enables the safe application of this cell line in many laboratories and deposition in suitable cell line banks. Nevertheless, as recommended for all other established cell lines, regular testing for inter- and intra-specific cross-contamination and bacterial impurities is still highly recommended to prevent falsified research results, misleading publications, and waste of research money [68].

The HSC lines HSC-T6 and CFSC-2G were previously characterized by our research team [19,20]. In the most recent study, important characteristics of a third rat HSC line (i.e., PAV-1) have now been described. In sum, each cell line has typical features, which makes them distinct from each other. Furthermore, the gene expression repertoire of each cell line might predict them for different research applications (Table 7).

## 5. Conclusions

In sum, the biochemical characteristics and the genetic features described in this study are suitable means for the identification of PAV-1 cells. Together with our recent publications in which our team reported unique elements for the rat HSC lines HSC-T6 and CFSC-2G, there are now three continuously growing HSC lines available with established genetic hallmarks allowing discrimination from each other. As such, the scientific value of results established with these cell lines will increase the biochemical research results and help to fulfill the demands of scientific journals and funding agencies when working with continuously growing cell lines. Nevertheless, one should always be aware that these continuous cell lines represent an immortalized model, and these may show different results compared to primary cells. The additional use of isolated primary cells should be critically evaluated depending on the scientific question.

## Figures and Tables

**Figure 1 cells-12-01603-f001:**
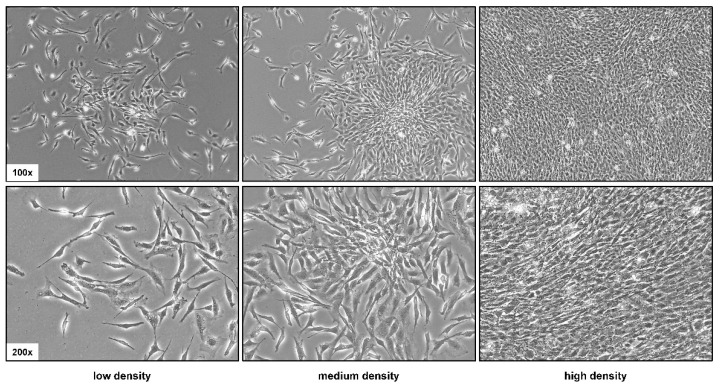
Phenotypic characteristics of PAV-1 cells. PAV-1 cells were seeded in cell culture dishes and representative images were taken at different densities. Original magnifications are 100× or 200×.

**Figure 2 cells-12-01603-f002:**
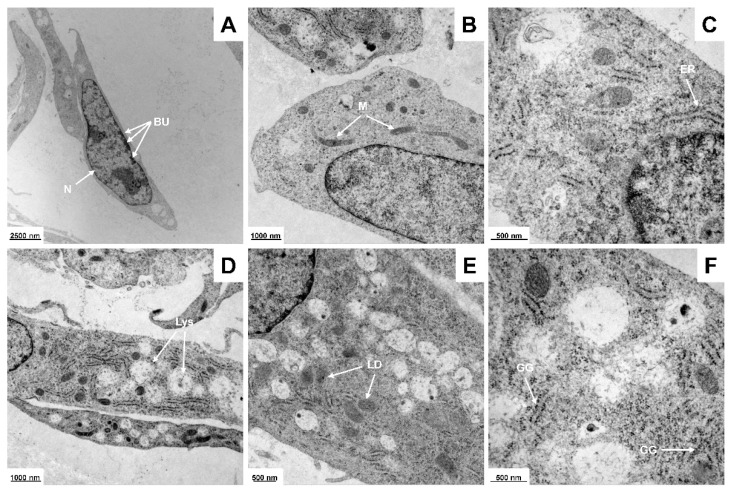
Electron microscopy of PAV-1 cells. Representative ultrastructural features of PAV-1 cells are shown, including the (**A**) large, electron-dense nucleus (N) with excessive bulges of the nuclear envelope (BU), (**B**) elongated mitochondria (M), (**C**) distinct rough endoplasmic reticulum (ER), (**D**) pronounced lysosomes (Lys), (**E**) abundance of intracellular lipid droplets (LD), and (**F**) the typical colloidal spherical glycogen granules (GG). Magnifications are (**A**) 4646×, (**B**,**D**) 10,000×, (**C**,**F**) 27,800×, and (**E**) 16,700×, respectively.

**Figure 3 cells-12-01603-f003:**
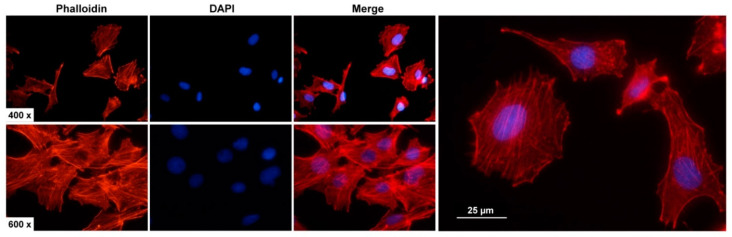
F-actin cytoskeleton staining in PAV-1. Cultured PAV-1 cells were stained with a Rhodamine-Phalloidin conjugate (red) and nuclei counterstained with DAPI (blue). Images were taken with a Nikon Eclipse E80i fluorescence microscope at 400× or 600× magnification. As previously shown, PAV-1 cells express large quantities of α-smooth muscle actin (*Acta2*) [23].

**Figure 4 cells-12-01603-f004:**
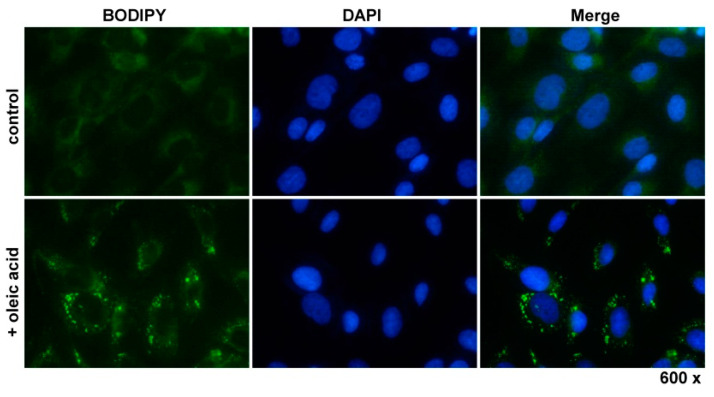
Lipid droplet stain. PAV-1 cells were treated for 24 h with 250 µM oleic acid or left untreated. Thereafter, the cells were fixed in paraformaldehyde, permeabilized, and cytoplasmic lipid droplets stained using the BODIPY dye.

**Figure 5 cells-12-01603-f005:**
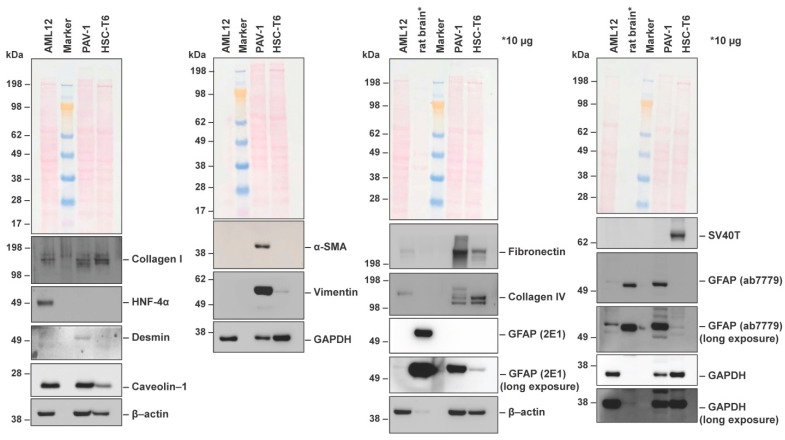
Hepatic stellate cell marker proteins in PAV-1 cells. Cell protein extracts were prepared from AML12, PAV-1, and HSC-T6 cells and analyzed by Western blot analysis (40 µg protein/lane) for expression of collagen type I, hepatocyte nuclear factor 1α (HNF-4α), desmin, caveolin-1, α-smooth muscle actin (α-SMA), vimentin, fibronectin, collagen type IV, glial fibrillary acidic protein (GFAP), and Simian virus (SV40) large T antigen (SV40T). Ponceau S staining and probing with antibodies specific for β-actin and glyceroaldehyde 3-phosphate dehydrogenase (GAPDH) served as controls to document equal protein loading. In this analysis, the cell line AML12 was used as a positive control for HNF-4α expression. * As a further control for GFAP expression, extracts prepared from rat brain (10 µg protein/lane) served as a positive control.

**Figure 6 cells-12-01603-f006:**
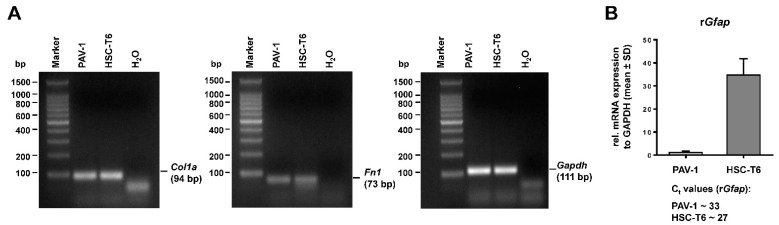
Conventional reverse transcription polymerase chain reaction (RT-PCR) and real-time quantitative PCR (qRT-PCR) for hepatic stellate cell markers. (**A**) Total RNA was isolated from PAV-1 and HSC-T6 cells and subjected to RT-PCR for expression analysis of collagen type I (*Col1a*), fibronectin-1 (*Fn1*), and glyceraldehyde 3-phosphate dehydrogenase (*Gapdh*) as a control to demonstrate the integrity of cDNA. (**B**) The expression of the glial fibrillary acidic protein (*Gfap*) in PAV-1 and HSC-T6 cells was analyzed by using qRT-PCR.

**Figure 7 cells-12-01603-f007:**
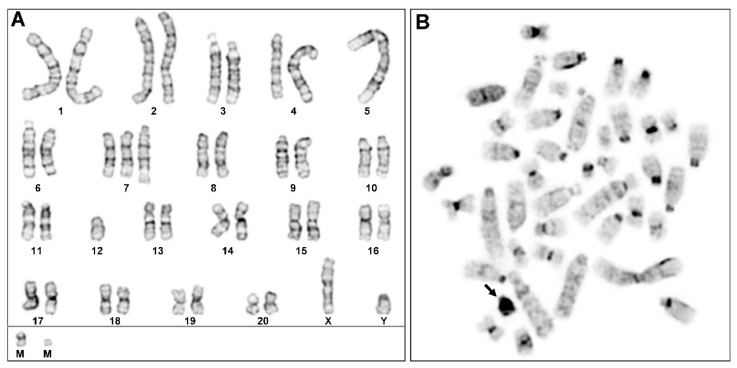
Conventional cytogenetic analysis of rat PAV-1 cells. (**A**) GTG-banded karyotype displaying structural and numerical rearrangements of specific chromosomes. At the bottom are the two small marker chromosomes (M). (**B**) CBG-banded metaphase spread showing the heterochromatic Y chromosome (marked by arrow).

**Figure 8 cells-12-01603-f008:**
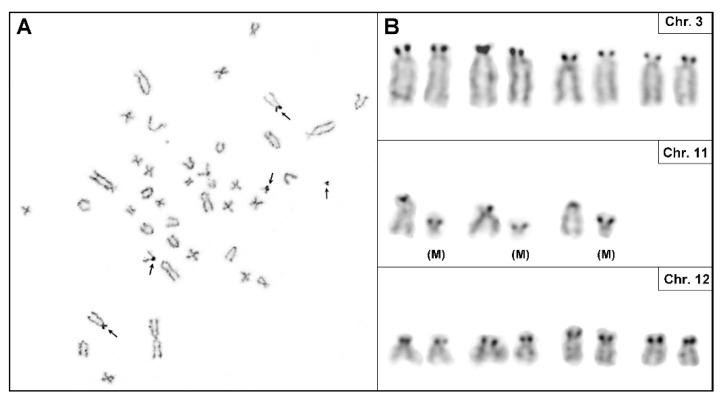
Visualization of nucleolar organizer sites (NORs) by silver staining. Silver-staining of metaphase spreads (**A**) as well as cut-outs of chromosomes 3, 11, and 12 (**B**) showing location of active NORs. Note the active NORs on marker chromosome (M) in all metaphases. Arrows in (**A**) indicate active NORs on chromosomes.

**Figure 9 cells-12-01603-f009:**
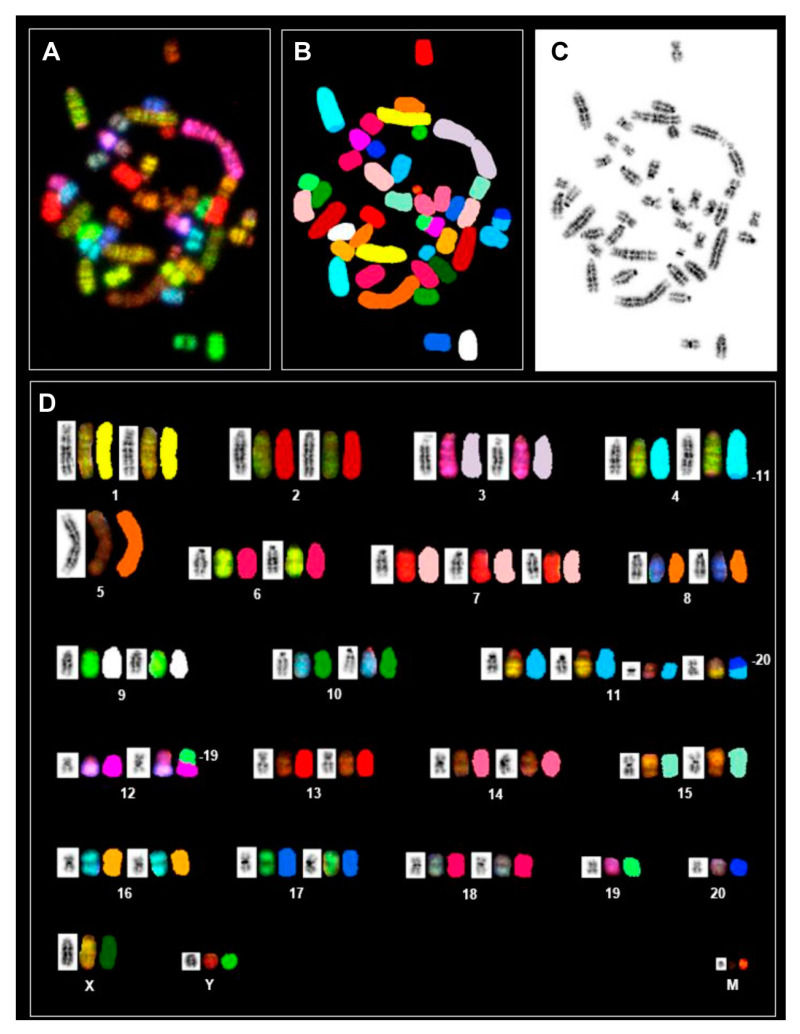
Spectral karyotyping of a rat PAV-1 cell metaphase. (**A**) RGB image after hybridization with the SKY probe cocktail that contains specific colored probes for each chromosome (**B**) Classified pseudo-colored image of the metaphase spread after hybridization with SKY paints. (**C**) Inverted DAPI-stained image. (**D**) Karyotype of the metaphase showing spectrally classified, pseudo-colored chromosomes (right) compared with its inverted DAPI-stained chromosomes (left) and corresponding RGB image (middle). Specific chromosomes involved in the rearranged chromosomes are indicated. Based on the SKY analysis, the PAV-1 karyotype can be designated as: 43, XY, der(4)t(4;11), i(5)(q10), +7, +der(11)t(11;20)+11(p), t(12;19), +mar.

**Table 1 cells-12-01603-t001:** Antibodies used for Western blot analysis.

Antibody	Cat. No.	Company	Expected Size (kDa)	Dilution	Clonality
α-SMA	CBL171-I	Sigma-Aldrich, Taufkirchen, Germany	45	1:1000	r mAb
β-actin	A5441	Sigma-Aldrich	43	1:10,000	m mAb
Caveolin-1	#3238	Cell Signaling; Leiden, The Netherlands	21–24	1:1000	r pAb
Collagen I	NB600-408	Novus Biologicals, Wiesbaden, Germany	139	1:1000	r mAb
Collagen IV	ab6586	Abcam, Berlin, Germany	161	1:1000	r pAb
Desmin (D93F5)	#5332	Cell Signaling	53	1:1000	r mAb
Fibronectin	AB1954	Sigma-Aldrich	262	1:3000	r pAb
GAPDH (6C5)	sc-32233	Santa Cruz Biotech., Santa Cruz, CA, USA	37	1:1000	m mAb
GFAP (2E1)	sc-33673	Santa Cruz	50	1:1000	m mAb
GFAP (astrocyte marker)	ab7779	Abcam	50 (55)	1:1000	r pAb
HNF-4α (C-19)	sc-6556	Santa Cruz	54 (40)	1:750	p gAb
SV40T (v-300)	sc-20800	Santa Cruz	94	1:1000	r pAb
Vimentin	ab92547	Abcam	54	1:3000	r mAb
g anti-mIgG (H + L), HRP	#31430	Invitrogen, ThermoFisher, Dreieich, Germany	NA	1:5000	g pAb
g anti-rIgG (H + L), HRP	#31460	Invitrogen	NA	1:5000	g pAb
m anti-gIgG (H + L) HRP	#31400	Invitrogen	NA	1:5000	m pAb

Abbreviations used are: g, goat; m, mouse; r, rabbit; IgG, immunoglobulin G; mAb, monoclonal antibody; pAb, polyclonal antibody; NA, not applicable.

**Table 2 cells-12-01603-t002:** Comparative gene expression in PAV-1, HSC-T6, and CFSC-2G ^1^.

Transcript ID	Gene ID	Gene	Gene Description	PAV-1 (TPM)	HSC-T6 (TPM)	CFSC-2G (TPM)
ENSRNOT00000083468.1	ENSRNOG00000058039.1	*Acta2*	actin alpha 2, smooth muscle	58.2526	2.377	10.9245
ENSRNOT00000005311.6	ENSRNOG00000003897.6	*Col1a1*	collagen type I alpha 1 chain	2470.3	2033.74	1408.42
ENSRNOT00000004956.4	ENSRNOG00000003357.4	*Col3a1*	collagen type III alpha 1 chain	423.594	2128.9	64.4932
ENSRNOT00000024430.5	ENSRNOG00000018087.5	*Vim*	vimentin	7038.71	2054.81	3853.69
ENSRNOT00000057585.4	ENSRNOG00000014288.8	*Fn1*	fibronectin 1	1693.83	143.953	1383.81
ENSRNOT00000019772.6				370.187	352.585	702.835
ENSRNOT00000017486.7	ENSRNOG00000012840.7	*Sparc*	secreted protein acidic and cysteine rich	2414.09	2188.34	1040.23
ENSRNOT00000067011.2	ENSRNOG00000003772.7	*Csrp2*	cysteine and glycine-rich protein 2	276.038	195.244	62.7395
ENSRNOT00000080598.1				4.09374	4.42437	1.12115
ENSRNOT00000013745.7	ENSRNOG00000010208.7	*Timp1*	tissue inhibitor of metallopeptidase 1	646.539	5518.57	714.162
ENSRNOT00000010180.5	ENSRNOG00000007650.5	*Cd63*	Cd63 molecule	2524.99	1639.48	2732.07
ENSRNOT00000090381.1				85.4183	60.6936	106.408
ENSRNOT00000011208.7	ENSRNOG00000008301.7	*Tagln2*	transgelin 2	1999.07	754.93	642.605
ENSRNOT00000034401.5	ENSRNOG00000002919.9	*Gfap*	glial fibrillary acidic protein	0.0115619	0.60103	0.108417
ENSRNOT00000093266.1				0	0	0
ENSRNOT00000093167.1				0	0	0
ENSRNOT00000042459.4	ENSRNOG00000034254.4	*Actb*	actin, beta	3767.55	4379.5	3869.05
ENSRNOT00000080216.1				1524.6	1820.7	1503.75
ENSRNOT00000050443.4	ENSRNOG00000018630.7	*LOC108351137*	glyceraldehyde-3-phosphate dehydrogenase	2100.85	4598.86	3223.93
ENSRNOT00000041328.3	ENSRNOG00000030963.3	*LOC108351137*	glyceraldehyde-3-phosphate dehydrogenase	1893.33	4243.2	2962.7

^1^ Expression values for HSC-T6 and CFSC-2G were taken from our previously published studies [19,20].

**Table 3 cells-12-01603-t003:** Expression of metabolic and nuclear retinoic acid receptors in PAV-1 in PAV-1, HSC-T6, and CFSC-2G ^1^.

Transcript ID	Gene ID	Gene	Gene Description	PAV-1 (TPM)	HSC-T6 (TPM)	CFSC-2G (TPM)
ENSRNOT00000008659.4	ENSRNOG00000009972.7	*RARα*	retinoic acid receptor, alpha	34.5437	24.8697	26.5339
ENSRNOT00000084644.1				25.8309	23.8544	20.4423
ENSRNOT00000033048.6	ENSRNOG00000024061.7	*RARβ*	retinoic acid receptor, beta	0.102317	0.541197	0.00872314
ENSRNOT00000016801.5	ENSRNOG00000012499.7	*RARγ*	retinoic acid receptor, gamma	67.7163	64.5919	49.0507
ENSRNOT00000017096.7				17.0844	39.1824	25.6328
ENSRNOT00000012892.4	ENSRNOG00000009446.4	*RXRα*	retinoid X receptor alpha	75.5966	18.9052	41.2102
ENSRNOT00000041613.5	ENSRNOG00000000464.7	*RXRβ*	retinoid X receptor beta	9.88301	20.0384	15.9738
ENSRNOT00000087670.1				1.0449	1.79029	1.53037
ENSRNOT00000086978.1				0.43314	1.08069	1.02833
ENSRNOT00000081588.1				2.00124	0.804912	1.61434
ENSRNOT00000091182.1				1.43022	0.723898	0.347194
ENSRNOT00000087895.1				0.824919	0.688452	0.0953106
ENSRNOT00000084638.1				1.03353	0.499241	0.615142
ENSRNOT00000079967.1				0.197368	0	0
ENSRNOT00000077227.1	ENSRNOG00000004537.6	*RXRγ*	retinoid X receptor gamma	0	0	0
ENSRNOT00000006117.5				0	0	0.271271
ENSRNOT00000018622.4	ENSRNOG00000013794.4	*Rbp1 (CRBP1)*	retinol binding protein 1	1.28709	25.3842	0
ENSRNOT00000018755.6	ENSRNOG00000013932.6	*Rbp2 (CRBP2)*	retinol binding protein 2	0.695375	0.309974	1.64899
ENSRNOT00000081756.1	ENSRNOG00000053850.1	*Rdh5 (Rdh1)*	retinol dehydrogenase 5 (formely 1)	0.65424	0.0383589	0.693011
ENSRNOT00000024000.6	ENSRNOG00000017619.7	*ALDH1A1*	retinal dehydrogenase 1	0.00983515	0.291398	0.0251453
ENSRNOT00000079115.1	ENSRNOG00000055049.1	*ALDH1A2*	retinal dehydrogenase 2	0.0287604	15.2533	0
ENSRNOT00000003182.5	ENSRNOG00000002331.5	*ALDH3A1*	aldehyde dehydrogenase 3 (Tumor ALDH)	240.833	21.5281	90.6901
ENSRNOT00000001816.6	ENSRNOG00000001344.7	*ALDH2*	mitochondrial aldehyde dehydrogenase	522.001	528.489	76.4596
ENSRNOT00000077461.1	ENSRNOG00000002342.6	*ALDH3A2*	microsomal ALDH	18.5197	8.59967	9.78843
ENSRNOT00000066109.3				73.9895	37.999	42.67

^1^ Expression values for HSC-T6 and CFSC-2G were taken from our previously published studies [19,20].

**Table 4 cells-12-01603-t004:** Number of diploid and tetraploid cells observed during serial cultivation of PAV-1 cells.

Passage No.	<Diploid	Around Diploid	Around Tetraploid
5th	1	83	17
10th	0	54	46
15th	1	39	61
20th	1	31	69
30th	2	39	59

**Table 5 cells-12-01603-t005:** Common and sporadic structural chromosomal rearrangements in PAV-1 cells identified through spectral karyotyping (n = 20).

Cell	Chromosome Numbers	der(1)t(1;3)	der(1)t(1;18)	der(2)t(2;11p)	der(3)t(3;12)	der(4)t(4;6)	der(4)t(4;11p)	der(5)t(5;5)	der(6)t(6;11p)	der(6)t(6;12)	der(7)t(7;16)	der(7)t(7;17)	der(7)t(7;19)	der(10)t(10;13/14)	der(10)t(10;19)	der(11)t(11;17)	der(11)t(11;18)	der(11)t(11;20)	der(12)t(12;16)	der(12)t(12;18)	der(12)t(12;19)	der(12)t(12;20)	der(16)t(16;17)	der(16)t(16;19)	der(17)t(17;18)
1	59,X			+				+										++			++				
2	80,XXYY	+					++	++								+	+				++			+	
3	77,XYYY			+			+	++										++			++		+		
4	39,XY				+		+	+										+							
5	41,XY						++	+										+			+				
6	77,XYY			+			++	++										++				+	+		
7	83,XXYY	+					++	++								+		++			++				
8	86,XXYY			+	++		++	++				+			+			++			+	+			+
9	80,XXYY	+					++	++								+	+	++		+					
10	69,XXYY			+			++	+	+									++			++		+		
11	83,XXYY	+					++	++								+		++			++				
12	43,XY						+	+										+			+				
13	81,XXYY	+					++	++								+		++			++				
14	43,XY						+	+										+			+				
15	80,XXYY	+						++								+		++			++				
16	83,XXYY			+			++	++		+								++			++		+		
17	81,XXYY			++			++	++			+			+				++			++		+		
18	84,XXY			+			++	++										++	+		++		+		
19	41,XY					+	+	+					+					+	+						
20	79,XXYYY	+	+					++								+		+			++				

+ rearrangements marked one time; ++ rearrangement marked twice mostly in tetraploid cells.

**Table 6 cells-12-01603-t006:** Comparison of STR profiles from PAV-1, CFSC-2G, and HSC-T6 using the 31 species-specific STR markers ^1^.

SN	Marker Name	Chromosomal Location	Allele Sizes (bp) in PAV-1	Allele Sizes (bp) in CFSC-2G	Allele Sizes (bp) in HSC-T6
1	73	1	194	194, 203	194
2	8	2	234, 238	236	234
3	2	2	128	126	127
4	4	3	236, 238	268, 270	238
5	3	3	162	160, 182	160, 162
6	26	4	154	150	166
7	19	4	179	180	175
8	81	5	130	130, 134	130, 132
9	34	6	182, 187	184, 189	188
10	30	7	192	188, 192	186, 192
11	24	8	254, 259	260	249, 253
12	59	9	145, 148	145	143, 146, 180
13	62	9	166	166	177
14	1	10	96, 105	105	96
15	55	10	210, 218	210, 214	210, 218
16	36	11	222	222	234
17	67	11	165	154, 156	165
18	13	12	121	121	121, 135
19	35	13	203	197	197, 203
20	42	13	144, 156	125	127
21	70	14	158, 175	158, 175	175, 179
22	61	15	128	128	128
23	79	15	172	172, 180	172
24	90	16	175	159, 161	174
25	69	16	136, 139	138	139
26	78	17	147, 149	136, 151	147, 151
27	15	18	232	232	232
28	16	18	251	251, 260	247, 251
29	75	19	144, 184	144	144, 184
30	96	20	210	210	210, 212
31	91	20	211, 225	221	205, 211

^1^ The STR profile of HSC-T6 and CFSC-2G were taken from our previous publications [19,20].

**Table 7 cells-12-01603-t007:** Comparison of cellular characteristics of rat hepatic stellate cell lines HSC-T6, CFSC-2G, and PAV-1 ^1^.

Characteristic	HSC-T6	CFSC-2G	PAV-1
Origin	Sprague Dawley	Wistar	Wistar
Sex	female ^2^	male	male
Immortalization	SV40T	cirrhotic liver (CCl_4_)	spontaneous
Morphology	fibroblastic (spindle shaped)	fibroblastic (flat)	fibroblastic (spindle shaped)
Chromosome number	<43>	<62>	<43>
Derivative chromosomes	der (1), der (4), der (7)	der(1), der(2), der(4), der(6), der(7) der(10), der(14), der(17), der(19)	der (4), der (5), der (11), der (12)
Additional chromosomes	+4, +7	+1, +2, +3, +4, +5, +6, +7, +8, +10 +13, +14, +17, +19	+7, +11, +mar
Missing chromosomes	none	−15, −16	none
STR profile	available [19]	available [20]	this study
Common HSC markers	see Table 2	see Table 2	see Table 2
SV40T expression	positive	negative	negative
Special medium requirements	none	non-essential amino acids	none
Proposed field of study	general HSC aspects retinoid metabolism	general HSC aspects collagen expression and biology	fat metabolism retinol metabolism

^1^ Note: (+) indicates the presence of multiple copies of specific chromosomes, while (−) indicates a deletion of a specific chromosome. SKY analysis was performed on chromosomes of the 15th passage; ^2^ originally reported to be established from a male rat, it was later shown that the cell line has a female karyotype [19].

## Data Availability

This manuscript contains most of the original data generated during our study. However, additional datasets (e.g., fastq data files of NGS analysis, additional illustrations of SKY painting) and results of repetitions of individual experiments are available upon reasonable request from the corresponding authors.

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
