# Peer review of "Genetic Characterization of Rat Hepatic Stellate Cell Line PAV-1"

_cells, 2023, doi:10.3390/cells12121603_

Round 1
Reviewer 1 Report
Verification of a novel immortalized cell line for research and clinical use is critical biomedical research. Cell line identification errors and cross-contamination are the most serious problems that trigger error-filled publications, faulty data, and non-reproducible results. In this manuscript, the authors use a serials analysis methods including Electron Microscopic Analysis, Metaphase Chromosomes and Karyotyping, STR profiling, Next Generation Sequencing, i.e., try to define unique authentication standards for PAV-1 cells that will be the basis to avoid misidentification when using this cell line in biomedical research. Overall, this is an interesting and substantial study. However, there are still several issues that need to be clarified by the authors.
1. Please clarify why the AML12 cell line was used as a control in the study instead of primary HSC.
2. In the past the authors have established genetic details of different HSC lines including and rat lines HSC-T6 and CFSC-2G. in this manuscript, it would be great if they can provide a detailed comparison (morphology, biomarker, Chromosomes & Karyotyping, and preferred field of study i.e.) between these three cell lines.
3. Both Figure 6 and Table 2 show that Gfap expression is more than 60-fold higher in the hematopoietic stem cell-T6 cell line than in the PAV-1 cell line, but this is inconsistent with the Western blot data in Figure 5, which indicates that GFAP expression is higher in the PAV-1 cell line than in the HSC cell-T6 cell line.
4. In Figure 6 A, no Gfap bands are shown due to the low level of Gfap expression in the cell line, so it is better not to use Figure 6A and just use Figure 6B.
5. The author should include the original PVA-1 cell line identification paper for the reference: PAV-1, a new rat hepatic stellate cell line converts retinol into retinoic acid, a process altered by ethanol. Int J Biochem Cell Biol. 2002 Aug;34(8):1017-29.
6. As the authors noted in their study, they used a mycoplasma-free PAV-1 cell line that was treated with their original mycoplasma-contaminated PAV-1 cell line by performing mycoplasma removal. The question is whether the clearance of mycoplasma can affect the properties of the original PAV-1 cell line.
7. Please give the full name/description of the abbreviation when it first been descripted.
Please check some formatting and typos in the manuscript.
Author Response
Dear Reviewer 1,
please find our response to your comments in the attached pdf-file.
Regards
Ralf Weiskirchen

Reviewer 2 Report
In this study, Gäberlein et al. conducted a comprehensive series of experiments, including karyotype analysis, bulk RNA sequencing, qRT-PCR and Western blot analysis, to characterize the genetic, cellular, and molecular properties of the rat hepatic stellate cell line PAV-1. The experiments were well design and executed, yielding the result of high quality. These findings establish a solid groundwork for utilizing PAV-1 cells as an in vitro model to investigate various biological aspects of hepatic stellate cells. A few issues should be addressed based on the study:
1. It would be beneficial to ascertain whether there are any available records regarding the passage history of the PAV-1 cell line when it was initially obtained from the original laboratory ten years ago. This information could offer valuable insights into the cellular character changes that might have occurred over time.
2. An intriguing observation made by the authors is the contrasting expression patterns of GFAP between PAV-1 and HSC-T6 cells. While the transcription level of GFAP is considerably lower in PAV-1 compared to HSC-T6, the protein expression level exhibits the opposite trend. To investigate this discrepancy, it is important for the authors to examine whether the qRT-PCR primers utilized for GFAP analysis can distinguish between the different transcription isoforms of GFAP. This analysis would help determine if the distinct transcriptional isoform profiles of GFAP exist in PAV-1 and HSC-T6 cells.
3. The authors employed conventional banding methods and SKY analysis to identify several unique chromosome structural rearrangements specific to the PAV-1 cell line. Considering that chromosome rearrangements increased in PAV-1 cells during passaging, it is essential for the authors to specify whether these features were identified in the early or late passages of the cell lines.
Author Response
Dear Reviewer 2,
please find our response to your comments in the attached pdf-file.
Regards
Ralf Weiskirchen

Round 2
Reviewer 1 Report
The manuscript is a marked improvement over the previous version in terms of writing, description of methods, and arrangement of data. The authors have responded appropriately to most of the issues raised by the reviewer. However, the review still concluded that for the analysis of all other immortalized hepatic stellate cell (HSC) lines, primary HSC should always be used as a baseline control.
Author Response
Dear Reviewer 1,
many thanks for your additonal comments. We have added short sentences in the text and conclusion to mark the relevance of primary hepatic stellate cells for verification of research findings in established hepatic stellate cell lines. In addition, a colleage of us has corrected several mistakes in grammar and spelling.
We hope you will agree that our work is now suitable for publication.
Regards
Ralf Weiskirchen